# SeekerGym: Benchmarking Agentic Information Seeking under Uncertainty

## Abstract

Effective information seeking is a prerequisite for AI agents, yet current systems often fail to autonomously identify, retrieve, and integrate relevant context. We propose SeekerGym, a modular environment for evaluating LLM agents on information-seeking tasks. Unlike prior benchmarks that focus on end-to-end task performance, SeekerGym evaluates agentic information seeking capabilities in two complex tasks: reconstructing Wikipedia pages and finding related literature for computer science survey papers. Furthermore, we design an information seeking agent called *SeekerAgent*, which employs various belief structuring pipelines including meta-reflection for cross-example learning. Through comprehensive experiments using SeekerGym, we evaluate several design choices for information seeking agents. We find that SeekerAgent improve recall by as much as 68% compared to frontier models.

## 1 Introduction

Large language models (LLMs) are increasingly used in complex agentic pipelines, such as deep research (Huang et al., 2025) and software development (Jimenez et al., 2024). A key part of many pipelines is information seeking (Xia et al., 2024), where agents must autonomously navigate complex environment to collect information useful for solving the downstream task. Information seeking is challenging since agents must act in an open world to discover available information, reasoning about their uncertainty to understand what information they may still need and how they might acquire it. It can be formulated as a Partially Observed Markov Decision Process (POMDP) (Tang et al., 2025), making it a challenging reinforcement learning problem. However, recent work shows that LLMs underperform on information seeking (Yang et al., 2024; Singh et al., 2025).

There has been substantial work on evaluating LLM agents, many of which require information seeking—e.g., in question answering (Mavi et al., 2024; Singh et al., 2025), the agent needs to retrieve relevant knowledge; in software development (Jimenez et al., 2024; Yang et al., 2024), the agent needs to retrieve relevant context; and in writing survey papers (Wang et al., 2024), the agent needs to find relevant research papers. However, existing benchmarks are end-to-end, making it difficult to assess whether failures are due to information seeking or some other part of the pipeline. Furthermore, information seeking failures can have severe consequences—missing information can lead to biases and misleading results that are difficult to identify. For instance, if a deep research agent fails to find conflicting experimental studies on a current research topic, it may incorrectly conclude that consensus has been reached; detecting such a failure is difficult since the user has no way of assessing the completeness of the surfaced studies.

We study the problem of evaluating and improving the information-seeking capabilities of LLM agents. Our contributions are threefold. First, we clarify and formalize the information-seeking task as a POMDP and introduce **SeekerGym**, a versatile POMDP environment for evaluating information-seeking capabilities. Second, we design **SeekerAgent**, a modular LLM agent that employs several belief structuring strategies including a meta-reflection approach and an uncertainty-augmented approach that provides substantial improvements. Third, we perform an extensive empirical evaluation on the information-seeking capabilities of frontier LLMs with different agentic designs, demonstrating that SeekerAgent can improve recall by as much as 68% compared to frontier models without belief structuring.

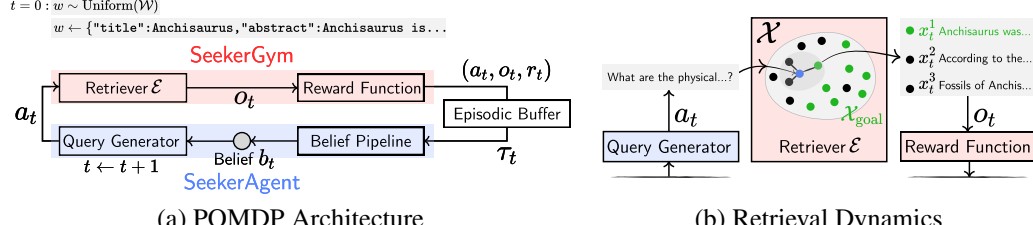

(a) POMDP Architecture      (b) Retrieval Dynamics

Figure 1: POMDP framework with four core components. (a) Our information-seeking architecture consists of SeekerGym (red components: retriever $\mathcal{E}$ and reward function $R$) and SeekerAgent (blue components: belief pipeline and query generator). SeekerGym tracks document coverage and retrieves passages from corpus $\mathcal{X}$; SeekerAgent maintains belief states and generates targeted queries to maximize information discovery. (b) Retrieval dynamics: Given a natural language query $a$ from the LLM agent, the retriever $\mathcal{E}$ samples the most relevant passages from corpus $\mathcal{X}$, returning observation $o$ that may contain both goal-relevant and non-goal passages.

## 2 SEEKERGYM: AN ENVIRONMENT FOR INFORMATION SEEKING

In this section, we formalize the information seeking task as a POMDP, and introduce the Seeker-Gym environment designed to evaluate information seeking capabilities of LLM agents.

### 2.1 POMDP FORMULATION

We consider an information-seeking task (Figure 1) where the agent is given a target topic (e.g., the title of a Wikipedia article in our Wikipedia environment, or the abstract of a survey paper in our survey paper environment), and their goal is to iteratively issue queries $a$ to an environment over a fixed corpus of passages $\mathcal{X} = \{x_1, \dots, x_N\}$ with the goal of uncovering a target subset of passages $\mathcal{X}_{\text{submit}} \subseteq \mathcal{X}$ within a certain number of iterations. With each query $a$, the agent receives an observation $o$ from the environment (e.g., the collection of passages retrieved from $\mathcal{X}$ by a fixed retriever $\mathcal{E}$) along with a reward signal $r$ encoding the amount of information uncovered.

Critically, the environment is not directly observable, so the agent must make decisions based on its observations. Thus, we formalize information seeking as a Partial Observable Markov Decision Process (POMDP). A (hidden) state $s \in \mathcal{S}$ encodes the environment (which is fixed) and the agent's progress (which is updated). Specifically, $s$ includes (1) the corpus of items $\mathcal{X} = \{x_1, \dots, x_N\}$; (2) a binary vector $s^{\text{goal}} \in \{0, 1\}^N$, where the $s_i^{\text{goal}}$ indicates whether $x_i$ it is relevant and must be retrieved; and (3) a binary vector $s_t^{\text{retrived}} \in \{0, 1\}^N$, where $s_{t,i}^{\text{retrieved}} = 1$ indicates whether the agent has already retrieved $x_i$. The action space $\mathcal{A}$ consists of all natural language queries, and the observation space $\Omega = 2^N$ encodes all subsets of items $\mathcal{X}$.

**Environments.** We provide three environments. Our CS surveys environment is the most complex. We curate a set $\mathcal{W}$ of computer science survey papers, and a set $\mathcal{X}$ of all research papers cited by some survey $w \in \mathcal{W}$. Papers are represented to the agent by their title and abstract. For the initial state, $\mathcal{X}$ is always the same, $s^{\text{goal}}$ is sampled by choosing $w \sim \text{Uniform}(\mathcal{W})$ and then taking $s^{\text{goal}} = \{x \in \mathcal{X} \mid w \text{ cites } x\}$, and $s^{\text{retrieved}} = \vec{0}$ is initially all zeros. Given a belief state representation $b \in \mathcal{B}$ (described in Section 3), an agent $\pi : \mathcal{B} \to \mathcal{A}$ outputs (possibly stochastic) actions in the form of natural language queries; this query is fed to a fixed retriever (e.g., from a vector store), which returns $k$ papers by their indices $\{i_1, ..., i_k\} \subseteq [N] = \{1, ..., N\}$ (where $k$ is a hyperparameter).

In addition, we prove two Wikipedia environments; they are designed to be conceptually similar but enable evaluating diverse distribution of information-seeking tasks. The two environments are identical except for the choice of $\mathcal{W}$ and $\mathcal{X}$. In both, $\mathcal{W}$ is a collection of Wikipedia articles and $\mathcal{X}$ is a set of passages in some article $w \in \mathcal{W}$; both are represented to the agent by their titles. As with the surveys environment, to construct an initial state, we choose $w \sim \text{Uniform}(\mathcal{W})$ and take $s^{\text{goal}} = \{x \in \mathcal{X} \mid w \text{ contains } x\}$ and $s^{\text{retrieved}} = \vec{0}$. The agent acts as before. We consider one benchmark with short articles (i.e., each article has a relatively small number of passages), and one with long articles.

Table 1: CS Surveys dataset

(a) Representative topic clusters

| Topic Cluster | Total Papers | For Evaluation |
|---|---|---|
| Wireless Communications | 357 | 10 |
| Medical Image Analysis | 317 | 10 |
| Graph Learning | 154 | 10 |
| Autonomous Driving | 147 | 10 |
| Federated Learning | 132 | 10 |
| *Other clusters* | *8,061* | 0 |
| **Total** | **9,168** | **50** |

(b) Hyperparameters

| Parameter | Value |
|---|---|
| *Content Filtering* | |
| Minimum abstract tokens | 48 |
| Maximum abstract tokens | 1024 |
| Available abstract count threshold | 75 |
| *Clustering Parameters* | |
| DBSCAN epsilon | 0.8 |
| DBSCAN min_samples | 128 |
| UMAP dimensions | 32 |

We provide additional details on how $\mathcal{W}$ and $\mathcal{X}$ are constructed in Section 2.2.

**Dynamics.** The observation function $O : \mathcal{S} \times \mathcal{A} \to \Omega$ implements a top-$k$ retrieval mechanism based on semantic similarity (i.e., retrieval from a vector store using vector embeddings):

$$O(s, a) = \text{top-k}_{x \in \mathcal{X}} \{\text{sim}(a, x)\}$$

where $\text{sim}(a, x)$ measures the semantic similarity between query $a$ and passage $x$. The initial state distribution $D$ is obtained by randomly sampling a goal vector, which occurs as described above, and initializing $s^{\text{retrieved}} = \vec{0}$; the corpus $\mathcal{X}$ is fixed. The transition function $T : \mathcal{S} \times \mathcal{A} \to \mathcal{S}$ is

$$s_{t+1,i}^{\text{retrieved}} = \begin{cases} 1 & \text{if } i \in o_t \\ s_{t,i}^{\text{retrieved}} & \text{otherwise,} \end{cases}$$

i.e., add $o_t$ to the list of retrieved items; the corpus $\mathcal{X}$ and goal $s^{\text{goal}}$ are fixed across time steps.

**Reward function.** We use a reward $r_t^{\text{info}}$ that provides a reward at each time step capturing the number of new target passages discovered at that step that are relevant:

$$R(a, s) = \sum_{i=1}^{N} s_i^{\text{goal}} \cdot (s_{t+1,i}^{\text{retrieved}} - s_{t,i}^{\text{retrieved}}).$$

**Train and test set.** Finally, we enable agents to learn across different problem instances. To this end, we can split the documents $\mathcal{W}$ into a training set $\mathcal{W}_{\text{train}}$ and a held-out test set $\mathcal{W}_{\text{test}}$. Then, the agent can be trained in the POMDP constructed using $\mathcal{W}_{\text{train}}$, and evaluated in the POMDP constructed using $\mathcal{W}_{\text{test}}$. The training set can be used both for traditional gradient-based training, as well as prompting-based learning strategies such as reflexion (Shinn et al., 2023).

## 2.2 DATASET CONSTRUCTION

**CS Surveys.** We collect computer science survey and review papers published between January 2024 and August 2025 from Semantic Scholar (Kinney et al., 2023), focusing on recent publications. To achieve balanced topic representation within computer science, we apply semantic clustering using embeddings of survey paper abstracts, grouping papers into coherent research areas through DBSCAN with UMAP dimensionality reduction. This clustering yields 11 distinct topics; Table 1a shows representative clusters demonstrating their diversity. This stratified sampling strategy prevents overrepresentation of dominant research trends while ensuring comprehensive coverage. Next, we retrieve cited research papers for each survey paper through the Semantic Scholar API, and filter papers based on various criteria. Table 1b shows hyperparameters for both filtering and clustering. Due to publisher restrictions, approximately 50% of cited papers had missing abstracts. After applying content filtering on the survey papers, we selected the top 10 papers from each cluster based on abstract availability ratio—ensuring minimal loss of original citation information from each survey.

**Wikipedia.** First, we collect all articles from the Wikipedia database (Wikimedia, 2024). To obtain a useful subset of high-quality articles, we first remove outlier and noisy articles, including those that are too sparse and contain predominantly fragmented short content (e.g., articles with many short bullet points). For sparse article removal, we leverage Wikipedia's hierarchical article

Table 2: Wikipedia dataset

(a) Representative topic clusters

| Topic Cluster | Total Articles | Ranked | For Evaluation |
|---|---|---|---|
| Video Games | 634 | → | 15 |
| MLB Players | 591 | → | 15 |
| Soap Opera Characters | 584 | → | 15 |
| Molecular Biology | 522 | → | 15 |
| Navy Ships | 519 | → | 15 |
| U.S. Highways | 437 | → | 15 |
| Islamic/Persian History | 361 | → | 15 |
| American Politicians | 357 | → | 15 |
| Chinese Dynasties | 356 | → | 15 |
| Bird Species | 325 | → | 15 |
| North American Geography | 242 | → | 15 |
| Railroads/Transit | 229 | → | 15 |
| Hotels and Real Estate | 205 | → | 15 |
| Dinosaurs | 203 | → | 15 |
| Romanian History | 159 | → | 15 |
| *Other clusters (19)* | *4,565* | | 0 |
| **Total** | **10,289** | | **225** |

(b) Hyperparameters

| Parameter | Value |
|---|---|
| *Outlier Removal* | |
| Minimum abstract tokens | 4 |
| Maximum abstract tokens | 1024 |
| Minimum passages | 8 |
| Minimum citations | 16 |
| Minimum references | 4 |
| *Sparse Article Removal* | |
| Minimum passage tokens | 40 |
| Sparse flag propagation threshold | 0.1 |
| Sparse ratio threshold | 0.1 |
| *Clustering Parameters* | |
| DBSCAN epsilon | 0.24 |
| DBSCAN min_samples | 128 |
| UMAP dimensions | 32 |

Table 3: Dataset statistics

| Dataset | Observations $|\mathcal{X}|$ | Observations per Document | | | Tokens per Observation | | |
|---|---|---|---|---|---|---|---|
| | | Mean±Std | (Min, Max) | Median | Mean±Std | (Min, Max) | Median |
| Short Wikipedia (SW) | 160K | 17.5±2.3 | (11, 25) | 17 | 159.6±87.7 | (6, 1332) | 144 |
| Long Wikipedia (LW) | 624K | 54.7±17.8 | (29, 143) | 53 | 148.1±81.3 | (4, 1767) | 132 |
| CS Surveys (CSS) | 384K | 135.9±47.7 | (77, 263) | 122 | 245.2±90.0 | (48, 1023) | 235 |

structure: starting from the deepest section level, we measure the ratio of content shorter than the minimum passage token threshold (40 tokens). When a section is detected as too short, its parent section considers this as sparse content. We recursively propagate this sparsity ratio up the hierarchy, filtering out articles with more than 10% sparse content at the root article level. Table 2b shows hyperparameters for both filtering and clustering. Second, we apply semantic clustering to group the remaining articles into topical clusters. Then, for each article, we perform paragraph-level segmentation to create passages $x \in \mathcal{X}$. Within each cluster, articles are ranked using a composite quality score that combines two factors: (1) article length, weighted by a bump function that favors medium-sized documents suitable for multi-turn exploration, and (2) citation counts as a proxy for article importance and completeness. The top 15 highest-scoring articles from each selected cluster constitute our evaluation set.

We create two variants of the Wikipedia dataset with different passage selection strategies. For Long Wikipedia (LW), we include all passages that survive the outlier removal and sparse article removal stages, resulting in 624K observations. For Short Wikipedia (SW), we apply an additional filtering step after clustering: passages are selected only from articles that pass clustering-based noise detection. This results in a smaller corpus of 160K observations but with higher content quality, as shown in Table 3.

**Statistics.** We show statistics for our datasets in Table 3, including the number of documents $|\mathcal{X}|$ ("Documents"), statistics on the number of observations per document (i.e., the number of items in $s^{\text{goal}}$) ("Observations per Documents"), and the number of tokens per observation (i.e., the number of tokens in an observation $x \in \mathcal{X}$) ("Tokens per Observation").

**Train/test split.** Rather than use a global train/test split, we perform the split at the level of clusters—i.e., we have a separate train/test split for each cluster. Then, we train the agent separately for each cluster, and test on the corresponding set. Results can be aggregated across all clusters. This strategy ensures the train and test sets encode related information, thereby facilitating learning.

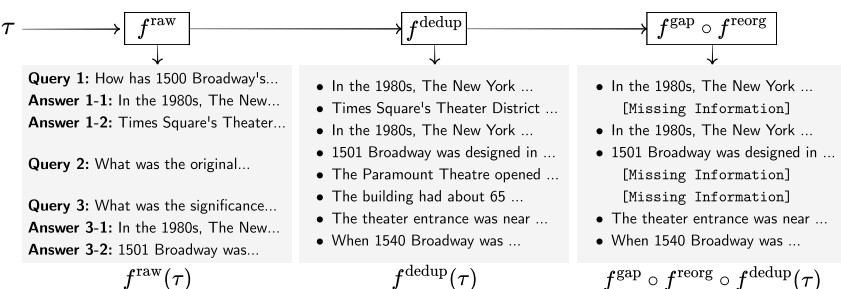

$$f^{\text{raw}}(\tau) \qquad\qquad f^{\text{dedup}}(\tau) \qquad\qquad f^{\text{gap}} \circ f^{\text{reorg}} \circ f^{\text{dedup}}(\tau)$$

Figure 2: Illustration of belief pipelines. Raw history preserves full dialogue but contains redundancy; deduplication removes duplicates; reorganization reorders passages; gap detection adds missing markers. These components refine the belief representation to support effective query generation.

# 3 SeekerAgent: Modular Agent Architecture with Compositional Belief Modeling

A SeekerGym agent has two components: (1) a *belief pipeline* that constructs a representation of the belief state, and (2) a *query generator* that generates an action based on the belief state.

## 3.1 Belief Pipeline via Modular Components

Performance in SeekerGym hinges on the representation $b \in \mathcal{B}$ fo the belief state—i.e., how the agent represents its belief state to guide future queries. The agent maintains an interaction history $\tau_t = \{(a_1, o_1), (a_2, o_2), ..., (a_t, o_t)\}$. However, naïvely representing beliefs to the agent as the raw interaction history $\tau_t$ leads to poor performance due to the its complexity. We consider various strategies for constructing effective belief state representations $b_t = f^{\text{belief}}(\tau_t)$. Intuitively, steps in this pipeline serve to identify knowledge gaps in the belief state and inspire more targeted queries. Below, we describe several pipelines, each of which are comprised of a sequence of reusable components. We visualize our pipelines in Figure 2, and provided a concrete example in Appendix D.3.

**Raw history.** This pipeline just uses the original interaction history $\tau_t$ as the belief state, including all queries and retrieved passages. While complete, this representation is verbose—e.g., it might be $b_t = [(a_1, \{A, B\}), (a_2, \varnothing), (a_3, \{A, C, D\})]$, including a repeated passage $A$ and failed query $a_2$.

**Deduplicated history.** This pipeline removes redundant information, returning all unique retrieved passages by the order of first appearance. For example, in the above example, after deduplicated history, the belief state would be $b_t = \{A, B, C, D\}$.

**Uncertainty-informed belief.** This pipeline first removes duplicated passages as before, but additionally reorganizes passages and tries to identify missing information. Specifically, it first implements a reorganization component $f^{\text{reorg}}$ that analyzes semantic relationships to reorder passages into a logical flow (e.g., $[C, D, A, B]$ if $C$ and $D$ are introductory). Second, it implements a gap detection component $f^{\text{gap}}$ that analyzes the reordered sequence and inserts explicit markers where content discontinuities are found (e.g., $[C, D, \texttt{[Missing]} \times 3, A, \texttt{[Missing]}, B]$). The marker $\texttt{[Missing]} \times 3$ indicate substantial missing content between $D$ and $A$, while the marker $\texttt{[Missing]}$ suggests a minor gap between $A$ and $B$.

**Meta-Reflection.** Finally, we consider a slight modification of the reflexion (Shinn et al., 2023) algorithm, which learns across problem instances by having an LLM "reflect" to summarize useful strategies. Specifically, we prompt an LLM to generate a summary of successful strategies deducible from same-cluster articles by viewing them as demonstrations to learn from. These strategies are appended to the belief pipeline prompt used for the corresponding test POMDP. In our evaluation, meta-reflection is used in conjunction with the deduplication history pipeline, but it can be combined with any pipeline.

**Prompts.** Prompt templates are shown in Appendices D.1 & D.2.

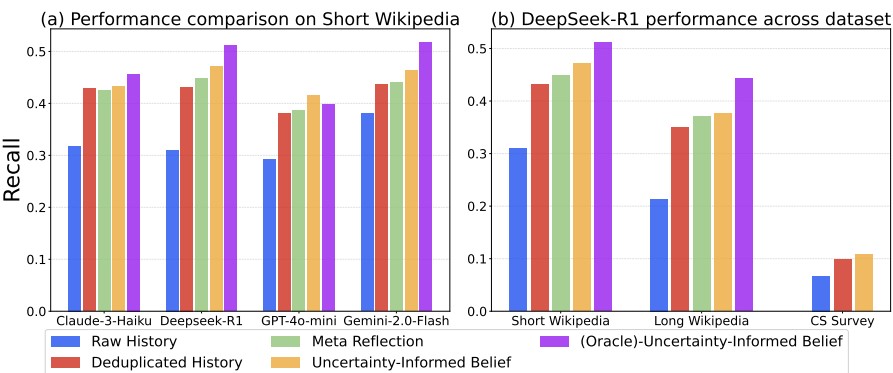

Figure 3: Comparative performance analysis across foundation models and belief structuring methodologies. (a) Performance comparison across LLM variants. The aggregated recall metrics demonstrate the relative efficacy of diverse large language model architectures (DeepSeek-R1, Gemini-2.0-Flash, GPT-4o-Mini, and Claude-3-Haiku) when evaluated on three benchmark datasets of increasing complexity: Short Wikipedia (SW), Long Wikipedia (LW), and Computer Science Surveys (CSS). Performance values represent the mean recall aggregated across all belief pipeline configurations for each model variant, providing a model-centric view of information retrieval capabilities. (b) DeepSeek-R1 performance stratified by belief pipeline configuration. The recall metrics illustrate the differential impact of four distinct belief structuring algorithms—Raw History, Deduplicated History, Uncertainty-Informed Belief, and Oracle-enhanced Uncertainty-Informed Belief—on information discovery effectiveness across the three dataset variants.

Table 4: Performance comparison across model variants and belief structuring methods on Wikipedia datasets (Short and Long Wikipedia combined). Left section shows recall values, right section shows average goals found per episode. Bold values indicate column maxima for each method, underlined values show the best non-oracle method for each model, and Gemini-2.0-Flash (bolded row) demonstrates overall superior performance.

| Model | Recall | | | | | Reward (Avg Goals Found) | | | | |
|---|---|---|---|---|---|---|---|---|---|---|
| | Raw | Dedup | Meta | UIB | Oracle | Raw | Dedup | Meta | UIB | Oracle |
| Claude-3-Haiku | 0.330 | **0.486** | 0.481 | 0.497 | 0.520 | 9.7 | **14.3** | 14.1 | 14.6 | 15.3 |
| Deepseek-R1 | 0.294 | 0.460 | **0.485** | 0.496 | 0.571 | 8.6 | 13.5 | **14.2** | 14.6 | 16.8 |
| GPT-4o-mini | 0.300 | 0.429 | 0.423 | 0.449 | 0.451 | 8.8 | 12.6 | 12.4 | 13.2 | 13.3 |
| **Gemini-2.0-Flash** | **0.396** | 0.471 | 0.479 | **0.508** | **0.605** | **11.6** | 13.8 | 14.1 | **14.9** | **17.8** |
| *Method Average* | 0.330 | 0.461 | 0.467 | 0.488 | 0.537 | 9.7 | 13.5 | 13.7 | 14.3 | 15.8 |

## 3.2 QUERY GENERATOR

The query generator takes the belief state $b_t$ and generates a set of new queries to continue exploration. It controls exploration strategy through parallel query generation. We set queries per turn to 10 by default to balance experimental efficiency with exploration coverage. The LLM backend selection affects both cost and capability; we tested multiple models with consistent prompting.

## 4 EXPERIMENTAL EVALUATION

We perform experiments to answer the following research questions: **RQ1:** Which foundation models excel at information seeking? **RQ2:** How do different belief pipelines affect information-seeking performance? **RQ3:** What is the cost-performance efficiency across different models and methods?

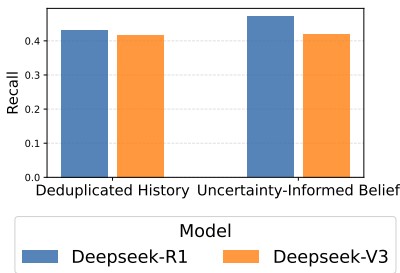

Figure 4: Ablation study comparing a reasoning model (DeepSeek-R1) to a similar non-reasoning model (DeepSeek-V3) across different belief pipeline configurations on our Short Wikipedia dataset.

---

**Example: Extended Reasoning Trace from Deepseek-R1**

```
''The ''Note'' says info was reorganized for coherence, and [Missing
Information] indicates gaps.  So maybe details about the IUCN listing
date, recent population estimates for the Common Pochard, or specific
conservation efforts beyond AEWA.
Possible gaps:  Current population trends for the Common Pochard.
The article mentions declines in Europe, but exact numbers?  Maybe
global population estimates.  Also, are there specific threats not
covered, like diseases or climate change?''
```

---

Figure 5: Qualitative example of reasoning in Deepseek-R1 when processing uncertainty templates. The model demonstrates extended reasoning by explicitly analyzing information gaps and generating specific hypotheses about missing details, including `[Missing Information]` markers.

## 4.1 EXPERIMENTAL SETUP

For SeekerGym, we use OpenAI's *text-embedding-3-large* embedding model (1536 dimensional vectors), with a similarity threshold being 0.7. We set the top-$k$ retrieval number to 10. For SeekerAgent, we use *Claude-3-Haiku*, *Deepseek-R1*, *GPT-4o-mini*, and *Gemini-2.0-Flash* as the LLM backends. We set the temperature of all models to 1.0, maximum query per iteration to 10, and maximum token length to 3200 for all models except Deepseek-R1, which uses 6400 tokens to accommodate its reasoning requirements.

**Experimental Configuration.** We conduct experiments across three datasets: Short Wikipedia (15 clusters, 15 episodes per cluster, 10 steps per episode), Long Wikipedia (15 clusters, 15 episodes per cluster, 15 steps per episode), and CS Surveys (5 clusters, 10 episodes per cluster, 15 steps per episode). Each episode is evaluated with a single run (no multiple trials per episode). We evaluate five belief pipeline configurations: (1) Raw History, (2) Deduplicated History, (3) Uncertainty-Informed Belief, (4) Meta-Reflection, and (5) Oracle-version of Uncertainty-Informed Belief. The last configuration replaces the LLM-based reorganization component $f^{\text{reorg}}$ and gap detection component $f^{\text{gap}}$ with oracles based on the ground truth; this variant is designed to represent the gap between LLM-based belief state representation and the best possible representation. Note that configurations (4) and (5) were not evaluated on the CSS survey task due to computational constraints from the extensive context length of survey papers, which limits the volume of data processable by the meta-reflection generator LLM, and due to unavailable Oracle information for survey paper infrastructure.

**Evaluation Metrics.** We measure retrieval performance using *recall*, defined as the fraction of relevant target paragraphs successfully retrieved by the agent out of all ground-truth paragraphs for a given task: recall $= |\mathcal{X}_{\text{retrieved}} \cap \mathcal{X}_{\text{goal}}|/|\mathcal{X}_{\text{goal}}|$. We focus on recall since our query generator outputs a fixed number of queries per iteration, making precision comparisons less informative across different belief pipelines. Additionally, we compute *cost* as the absolute LLM usage cost in USD, calculated using each provider's tokenizer with input token cost weights and output token cost weights (USD per million tokens), ensuring valid cost comparisons across different LLM variants. Detailed cost calculation methodology is provided in Appendix B.

Table 5: Efficiency ratios across model variants and belief structuring methods on Short and Long Wikipedia. Values represent $|\mathcal{X}_{\text{retrieved}} \cap \mathcal{X}_{\text{goal}}|/(1000 \times \text{total cost})$ , where higher values indicate better cost-effectiveness. Bold values show column maxima, underlined values show the best non-oracle method for each model.

| Model | Raw History | Dedup. History | Meta Reflection | UIB | Oracle-UIB |
|---|---|---|---|---|---|
| Claude-3-Haiku | 0.308 | 0.987 | 0.968 | 0.650 | 1.165 |
| Deepseek-R1 | 0.199 | 0.405 | 0.418 | 0.098 | 0.459 |
| GPT-4o-mini | 0.639 | 1.654 | 1.564 | 0.375 | 1.853 |
| **Gemini-2.0-Flash** | **1.109** | **2.417** | **2.267** | **1.306** | **3.243** |

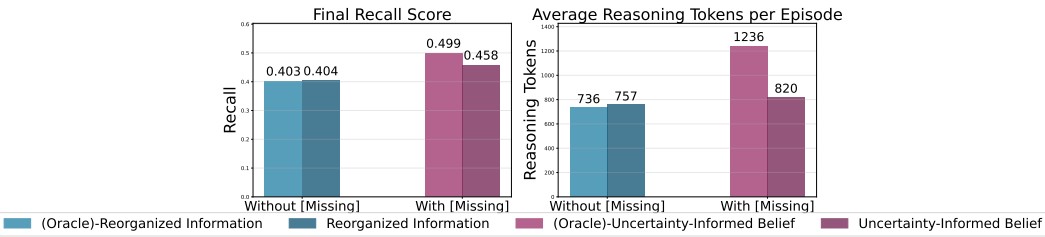

Figure 6: Ablation study analyzing the impact of the gap detection component $f^{\text{gap}}$ on performance and token usage. Specifically, it "Without Gaps" removes the gap detection component $f^{\text{gap}}$ from the Uncertainty-informed belief pipeline as well as from the oracle version of this pipeline.

## 4.2 RESULTS

**RQ1: Foundation models.** First, we assess different foundation models on information-seeking performance. This is crucial for understanding whether general model capabilities correlate with effective exploration or if specialized competencies are required; results are shown in Figure 3a and Table 4. As can be seen, Gemini-2.0-Flash generally outperforms other models.

A particular question of interest is the impact of reasoning on performance. In Figure 4, we show an ablation comparing Deepseek-R1 (a reasoning model) to Deepseek-V3 (its non-reasoning base model) for two of our pipelines on Short Wikipedia. As can be seen, reasoning improves performance. However, looking at the broader pattern in Table 4, we observe that other non-reasoning models (Gemini-2.0-Flash) achieve the best performance across most belief pipeline variants. We hypothesize that since reasoning models are typically trained for tasks like mathematical problem solving, they may not consistently excel at information seeking, which necessitates the design of specialized information seeking agents.

**RQ2: Belief structuring.** Next, we investigate the effect of our belief pipelines. The purpose of this experiment is to determine how progressively sophisticated belief representations impact exploration efficiency. By structuring the agent's belief state, we aim to reduce redundant searches and improve the generation of targeted queries. Results are shown in Figure 3b and Table 4. As can be seen, even deduplication yields a noticeable improvement over raw history. Meta-reflection achieves a small improvement over deduplication. Our uncertainty-informed belief pipeline, which explicitly models information gaps, achieves the highest performance by precisely targeting missing content. Figure 6 shows an ablation where we remove the gap detection component $f^{\text{gap}}$. Additionally, we find that the reasoning traces exhibit extensive belief structuring behaviors when they notice `[Missing]` markers; see Figure 5 for an example.

**RQ3: Cost-performance tradeoff.** We analyze the tradeoffs between computational cost and retrieval performance across different models and methods. Table 5 shows results. Our cost-performance analysis reveals that Gemini-2.0-Flash emerges as the most cost-efficient model, achieving competitive recall performance at significantly lower cost per episode compared to reasoning-heavy models like DeepSeek-R1. As can be seen in Table 4, while component $f^{\text{gap}}$ improves performance, it also significantly improves the number of reasoning tokens, leading to its comparatively worse performance, as can be seen in Figure 7. Among belief pipelines, Deduplicated History achieves the best cost-performance efficiency, due to its simple design of information duplication handling mechanism.

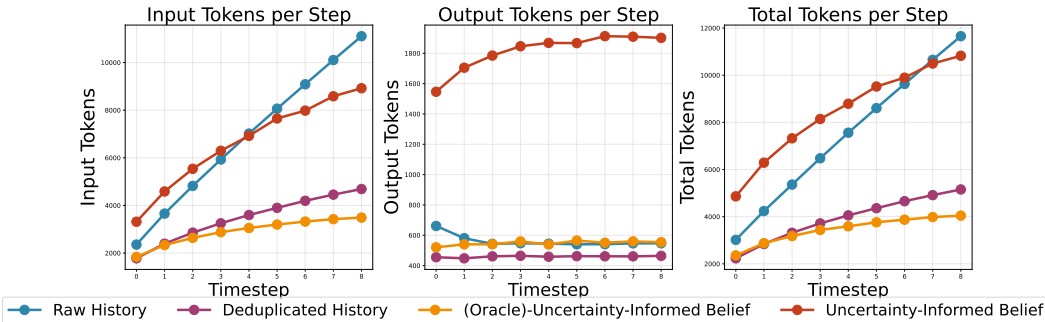

Figure 7: Token usage comparison across different methods and models. The analysis shows how different belief pipeline configurations affect computational costs through token consumption, providing insights into the efficiency trade-offs of various approaches.

## 5 RELATED WORK

**Information seeking.** Prior work has explored how to use deep RL to train information seeking agents, e.g., to iteratively search until finding an optimal information source (Narasimhan et al., 2016). More broadly, there has been work on algorithms for active information gathering in control (Sadigh et al., 2016), which motivates our POMDP formulation. However, their approach requires gradient-based training of the agent, which can be prohibitively expensive. In contrast, our belief representation pipeline improves information seeking without finetuning.

**Benchmarking LLM Agents.** While there are a large number of benchmarks for assessing the performance of LLM agents, they use end-to-end tasks instead of isolating the information seeking components. For instance, question-answering benchmarks such as HotpotQA (Yang et al., 2018) and MultihopQA (Mavi et al., 2024) require information seeking, but conflate these capabilities with reasoning and answer generation. Recent DeepResearch benchmarks (Du et al., 2025; Bosse et al., 2025) similarly focus on end-to-end generation of survey papers instead of isolating the task of identifying relevant research papers. SWE-Bench (Jimenez et al., 2024) similarly evaluate end-to-end performance, making it difficult to isolate information-seeking failures. SeekerGym addresses this gap by providing a benchmark specifically for evaluating information seeking.

## 6 CONCLUSION

We have introduced SeekerGym, a comprehensive benchmark for evaluating information-seeking capabilities of LLM agents in multi-turn scenarios. We frame the problem as a Partially Observable Markov Decision Process (POMDP), based on which we design the SeekerGym environment. Furthermore, we have designed SeekerAgent, which uses a compositional belief state representation pipeline that explicitly models both content structure and agent uncertainty to improve the performance of downstream query generation. Through extensive experiments, we have demonstrated that structured belief representations significantly improve information seeking capabilities. Information seeking forms a critical component of many agentic systems, and more work is needed to design effective information seeking agents.

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

## A    SEEKERGYM DETAILS

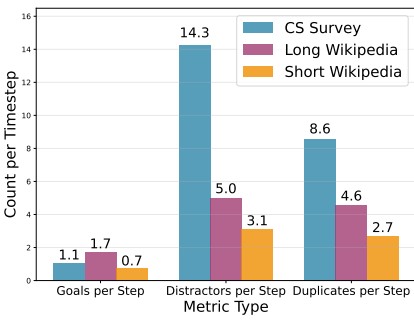
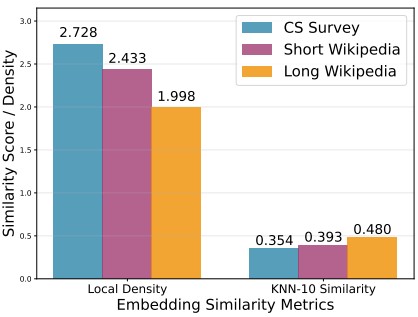

(a) Dataset Comparison: Per-Step Rates        (b) k-NN-based environment analysis

Figure 8: Dataset characteristics and environment analysis. (a) Per-step rates across three datasets showing unique goals retrieved per step from $|\mathcal{X}^{\text{goal}}|$, unique distractors accumulated per step, and duplicates encountered per step in $\tau_t$ that agent has as episodic memory. (b) k-NN based environment investigation measuring local distance patterns in each environment's embedding space $\mathcal{X}$ to analyze retrieval difficulty and information density.

## B    COST CALCULATION DETAILS

We compute experimental costs using model-specific tokenizers and official pricing rates. For each model, we use the appropriate tokenizer:

**Claude-3-Haiku**: o200k base tokenizer (due to API latency constraints)

**GPT-4o-mini**: o200k base tokenizer

**DeepSeek-R1**: Open-source V3 tokenizer

**Gemini-2.0-Flash**: Vertex AI tokenizer

Costs are calculated by multiplying token counts (input and output) by the respective model's pricing rate (USD per million tokens) as published by each provider.

## C    TOPIC-SPECIFIC PERFORMANCE INFORMATION

This section analyzes performance variations across different topic clusters, examining how domain-specific characteristics affect information-seeking effectiveness. We aggregate the metrics over all methods and LLM models to identify topic-level patterns. (Figure 9, 10)

## D    SEEKERAGENT DETAILS

### D.1    BELIEF PIPELINE COMPONENT PROMPTS

We employ unified prompt templates across all belief components to ensure consistency in query generation. The actual implementation uses the following prompt templates:

**Base Agent Introduction**

```
You are an elite, AI-Powered Search Agent tasked with
retrieving the most relevant information from a search
engine to complete an article about {topic}.
```

**Initial Exploration Prompt**

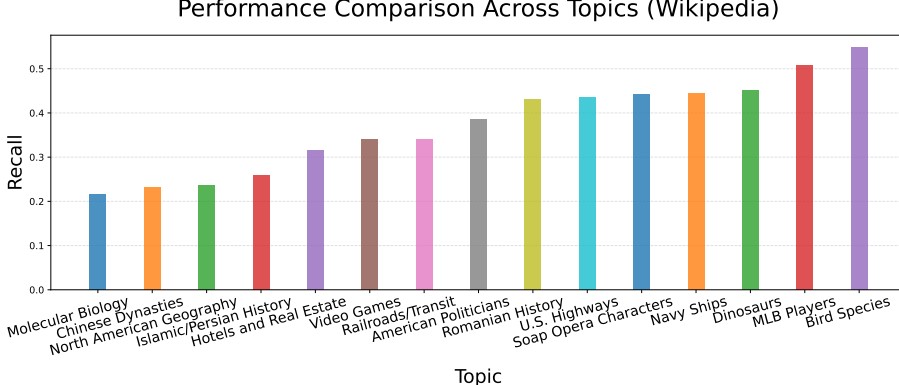

Figure 9: Performance across Wikipedia topic clusters. We aggregate recall metrics across all models (Claude-3-Haiku, Deepseek-R1, GPT-4o-mini, and Gemini-2.0-Flash) and belief pipeline methods to analyze topic-level performance patterns. Results show that LLMs struggle with information seeking in technical domains like Molecular Biology and Chinese Dynasties, which require specialized terminology and domain expertise, but perform well on casual topics like Bird Species and MLB Players, where the vocabulary and concepts are more accessible and commonly represented in training data.

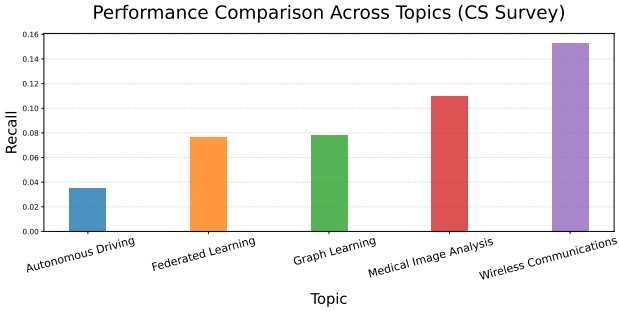

Figure 10: Performance across CS Surveys topic clusters. We aggregate recall metrics across all models (Claude-3-Haiku, Deepseek-R1, GPT-4o-mini, and Gemini-2.0-Flash) and belief pipeline methods to analyze topic-level performance patterns within computer science research domains.

```
{base_intro}

{abstract_section}No observations have been retrieved yet.
Your task is to formulate initial search queries to begin
exploring the topic comprehensively.

Instructions:
1. Generate broad, exploratory queries to understand
   different aspects of {topic}.
2. Wrap each question in:
   <Question>...your natural-language question...</Question>
3. Generate up to {num_queries} questions total.
```

**Standard Instructions Template**

```
{base_intro}

Your task is to analyze {content_description} and formulate
targeted follow-up search queries to fill any identified
```

```
information gaps.

Instructions:
1. {analysis_instruction}
2. Whenever you notice missing information, ambiguity,
   or anything that requires further information for
   article completion, pose a question to retrieve the
   corresponding information.
3. Generate natural language questions about the content
   itself. Do NOT include any metadata{metadata_note}
   in your questions.
4. Wrap each question in:
   <Question>...your natural-language question...</Question>
5. Generate up to {num_queries} questions total.
```

### D.2 META REFLECTION PROMPT

**Input Prompt: Topic**

```
Analyze the content patterns across these articles and create a
GENERALIZED information structure that captures how this topic is
typically organized.

YOUR TASK: Identify common information groupings that appear
across multiple articles. Create logical sections that would
apply to ANY article in this topic.

OUTPUT REQUIREMENTS: - Use keyword-focused, concise descriptions
- Avoid filler words like "This section contains..." or "includes
information about..." - List information types directly (e.g.,
"name, type, classification" NOT "This section introduces the
name, type, and classification") - Create 5-10 main sections
maximum - Use abstract, generalizable section names

OUTPUT FORMAT: <ANSWER> ## Section Name 1 keyword1, keyword2,
keyword3, brief-description-of-content-type

## Section Name 2 technical-specs, measurements,
performance-data, comparative-metrics

## Section Name 3 origins, development-timeline, key-events,
predecessor-information

(Continue for all major content groupings you identify)
</ANSWER>

IMPORTANT: - Be CONCISE and KEYWORD-FOCUSED in descriptions - NO
specific names, dates, or examples from the articles - Focus on
TYPES of information, not specific content - Each section should
be reusable for ANY new article in this topic
```

**Output Sample (Topic: Video Games)**

```
## Overview & Core Mechanics

gameplay-type, genre-classification, perspective-view,
control-scheme, primary-objectives, basic-rules

## Gameplay Systems & Features
```

```
combat-mechanics, progression-systems, multiplayer-modes,
difficulty-settings, special-abilities, customization-options

## Narrative & Setting

plot-summary, main-characters, world-description,
timeline-placement, thematic-elements, story-structure

## Development & Production development-team, creation-timeline,
technical-engine, design-philosophy, production-challenges,
platform-decisions

## Audio & Visual Design

graphics-style, art-direction, soundtrack-composition,
voice-acting, technical-specifications, aesthetic-influences

## Release & Distribution

launch-dates, platforms, regional-availability,
special-editions, downloadable-content, post-launch-updates

## Reception & Impact

critical-scores, review-highlights, sales-figures,
awards-recognition, cultural-influence, legacy-assessment

## Related Media & Franchise

sequels-prequels, spin-offs, adaptations, merchandise,
expanded-universe, series-connections
```

### D.3 BELIEF COMPONENT USAGE EXAMPLES

We provide a concrete example of how each belief pipeline transforms the interaction history.

**Raw history format $f^{\text{raw}}$.** The raw history preserves the complete dialogue:

```
Query 1: What are the main features of quantum computing?
Answer 1-1: Quantum computers use qubits instead of classical bits...
Answer 1-2: Superposition allows qubits to exist in multiple states...
Answer 1-3: Quantum entanglement enables correlated quantum states...

Query 2: Explain quantum error correction techniques
Answer 2-1: Error correction is crucial because quantum states are fragile...
Answer 2-2: Quantum computers use qubits instead of classical bits...
Answer 2-3: Surface codes are a popular error correction method...
```

Note the duplication: Answer 2-2 repeats Answer 1-1 verbatim, creating redundancy.

**Deduplicated observations $f^{\text{dedup}}$.** After deduplication, only unique passages remain:

```
[1] Quantum computers use qubits instead of classical bits...
[2] Superposition allows qubits to exist in multiple states...
[3] Quantum entanglement enables correlated quantum states...
[4] Error correction is crucial because quantum states are fragile...
[5] Surface codes are a popular error correction method...
```

**Reorganized with missing placeholders $f^{\text{gap}}$.** The final pipeline marks gaps:

```
[1] Quantum computers use qubits instead of classical bits...
[2] Superposition allows qubits to exist in multiple states...
[3] Quantum entanglement enables correlated quantum states...
[4] [Missing Information] [Missing Information] [Missing Information]
[5] Error correction is crucial because quantum states are fragile...
[6] [Missing Information]
[7] Surface codes are a popular error correction method...
```

The gap detection system identified that intermediate concepts (e.g., decoherence, noise models) are likely missing between basic quantum properties [1-3] and error correction [5], inserting three markers. A single marker at [6] suggests minor missing details about error correction approaches.

