# OpenReview forum: "SeekerGym: Benchmarking Agentic Information Seeking under Uncertainty"
_ICLR.cc/2026/Conference — ICLR 2026 Conference Withdrawn Submission_

### Official Review · Reviewer_c1Hi · 2025-10-15

**Soundness:** 2
**Presentation:** 1
**Contribution:** 2
**Rating:** 2
**Confidence:** 4

**Summary:**

This paper introduces SeekerGym, the authors formulate the task as a POMDP and provide curated Wikipedia and CS survey datasets, along with a comprehensive suite of belief-structuring pipelines for agent state representation. The modular SeekerAgent leverages various belief modeling approaches, including meta-reflection and explicit uncertainty modeling, and is evaluated extensively on the proposed benchmark. The experiments explore different LLM backbones, agent architectures, belief pipelines, and cost-performance tradeoffs, with detailed results and ablations.

**Strengths:**

- The paper identifies and fills an important gap in the empirical evaluation of agentic LLMs by systematically isolating information-seeking from confounding downstream tasks, which clear focus on isolating information-seeking abilities.
- The Wikipedia and computer science survey datasets are large, diverse, and constructed with explicit filtering and clustering strategies. (from description, and I haven't check the dataset.)
- Results are broken down by topic, agent design, and model, offering a deep diagnostic view.

**Weaknesses:**

- Writing related:
    - 1. Related work is incomplete: Several directly relevant prior works on agentic information seeking are not listed.
    - 2. some typo should be fixed.
    - 3. more detailed caption is needed (eg. Table 2); and some caption should be concise (eg. Figure 3)

- Lack of direct experimental comparison with existing benchmarks/methods: Despite detailed descriptions of the environment and agent pipeline, the paper lacks experiments against existing benchmarks, limiting assessment of SeekerGym's novelty and difficulty. It also provides no experimental results on how introdeced method perform on prior agentic benchmarks.

- Insufficient novelty: The method is well-motivated but largely follows established RL conventions. Most contributions are architectural and limited. The belief structuring pipelines are software engineering contributions more than theoretical ones.

**Questions:**

- Can the authors provide direct empirical comparisons against at least one or more prior agentic information-seeking benchmarks (e.g., WebArena, WebShop, GAIA, browsecomp-en/cn)? This would clarify SeekerGym's relative challenge and agent improvement.

- More experimental results of your agent and more models is needed (which is necessary for a benchmark) on other open-sourced benchmarks, not just SeekerGym

- More analysis on open-sourced/thinking/no-thinking models (not just api-based)

---

### Official Review · Reviewer_4MiK · 2025-10-31

**Soundness:** 2
**Presentation:** 1
**Contribution:** 2
**Rating:** 2
**Confidence:** 4

**Summary:**

This paper introduces SeekerGym, a modular environment designed to evaluate large language models (LLMs) on information retrieval tasks. It includes two types of tasks: finding related literature for Computer Science(CS) survey papers and reconstructing Wikipedia pages.  SeekerGym formalizes information retrieval as a Partially Observable Markov Decision Process (POMDP) and incorporates a SeekerAgent that employs various belief-structuring strategies, including meta-reflection and uncertainty-augmented approach methods, leading to a substantial improvement in retrieval recall.  The experiments evaluate the impact of different belief pipeline configurations on performance.

**Strengths:**

1.	Information retrieval has long been a key focus in the LLM community due to its strong application potential. Modeling this problem as a POMDP is an intuitive and reasonable approach.

2.	The two environments designed in this paper: finding related literature for CS survey papers and reconstructing Wikipedia pages are sufficiently long-horizon and complex, closely aligned with real-world scenarios. The pipeline is carefully designed with well-considered data cleaning and filtering procedures.

3.	The experimental section is organized around three key questions: which models demonstrate stronger information retrieval capabilities, how different belief pipelines affect model performance, and how to balance the cost–performance trade-off. The logic is relatively clear, and the analysis appropriately considers the constraints imposed by computational cost, adding practical significance to the study.

**Weaknesses:**

1.	The writing of this paper presents some issues. The methodological description is overly detailed, making the paper resemble a technical report rather than a research article. Some hyperparameters and implementation details could be moved from the main text to the appendix, and the formatting of figures(Figure 4) and tables(Table 1-3) could be further optimized.

2.	The overall workload appears limited. The paper presents an interactive system and explores the impact of different prompts on model information retrieval performance; however, the range of evaluated models is too narrow and focuses exclusively on closed-source models, which substantially weakens the validity and generalizability of the experimental conclusions. Conducting experiments on a wider range of models(such as Qwen3, GPT-5 and LLaMA-3.1) would make the conclusions more convincing and generalizable.

3.	The references are too few and outdated, with only 16 citations (including two websites). The Related Work section lacks discussion of existing benchmarks for LLM-based information retrieval, not limited to agent settings. In addition, there is a factual error in Line 455: Memento and UoT have already proposed prompt-based retrieval augmentation methods.

4.	Section 3.1 states that Meta-Reflection is an improved version of Reflexion, but the paper lacks comparative experiments against the baseline Reflexion method. Moreover, since Meta-Reflection combines Reflexion with Deduplicated History, its novelty appears limited. In addition, Line 268 claims that Meta-Reflection “can be combined with any pipeline”, yet no experiments are provided to substantiate the general applicability of Reflexion.


Minor Points:

1.	The caption of Figure 3 is excessively long, spanning 11 lines. Moving part of the caption’s content into the main text would improve the paper’s readability and conciseness.

2.	The paper does not clearly articulate the contribution and novelty of SeekerGym. Providing a comparison table highlighting the differences between SeekerGym and other existing benchmarks would substantially improve clarity and strengthen the presentation.

[1] Zhou H, Chen Y, Guo S, et al. Memento: Fine-tuning llm agents without fine-tuning llms[J]. arXiv preprint arXiv:2508.16153, 2025.

[2] Hu Z, Liu C, Feng X, et al. Uncertainty of thoughts: Uncertainty-aware planning enhances information seeking in large language models[J]. arXiv preprint arXiv:2402.03271, 2024.

**Questions:**

1．	In Line 355, all models are configured with a temperature of 1, and no independent repeated experiments are reported. This setup may introduce excessive randomness and could undermine the reliability of the conclusions.

2．	I am curious whether the performance of different models on the SeekerGym benchmark is consistent with their performance on other information-seeking benchmarks. If discrepancies exist, what factors contribute to these differences? Additionally, I would like to know whether SeekerAgent also demonstrates strong performance on other benchmarks.

3．	Including several case studies(both successful and failed) would help readers better understand and engage with the paper.  I would like to know whether the design of Meta-Reflection encourages more diverse response patterns from the model, thereby contributing to its improved performance.

---

### Official Review · Reviewer_Z2dW · 2025-10-31

**Soundness:** 3
**Presentation:** 3
**Contribution:** 2
**Rating:** 2
**Confidence:** 4

**Summary:**

- This paper introduces SeekerGym, a new benchmarking environment for evaluating information-seeking behavior in LLM agents. The authors formalize the task as a Partially Observable Markov Decision Process and provide three benchmark datasets—Short Wikipedia, Long Wikipedia, and Computer Science Surveys—to test agentic exploration and retrieval capabilities. They also propose SeekerAgent, a modular agent architecture incorporating Belief pipelines and Query generation modules.

**Strengths:**

- Rigorous Formalization: The POMDP formulation is mathematically sound and provides a principled framework for the information-seeking task with clear state/action/observation spaces.
  - Modular Design: The compositional belief pipeline approach is elegant and allows for systematic ablation studies to understand which components contribute to performance.

**Weaknesses:**

- Incomplete Paper: The paper mentions "we train the agent separately for each cluster, and test on the corresponding set" (line 214) and "Train and test set" (line 140), but the experimental section only uses existing APIs without any description or analysis of the training process.
  - Limited Novelty: The proposed belief pipeline essentially only manages memory and optimizes query generation; the feedback mechanism of the constructed SeekerGym is oversimplified compared to real web environments, which is insufficient to validate the claimed improvement in information seeking capabilities.
  - Missing Baseline Comparisons: No comparison with retrieval-augmented generation (RAG) baselines or specialized information retrieval methods.

**Questions:**

- Testing Methodology: During testing, do you test each topic (cluster) separately and then average the results? Why not perform seeking in one large dataset? Would this approach result in lower recall?
  - Interpretation of Figure 9: Does Figure 9 suggest that backend models may have different understanding of knowledge across professional domains due to training data differences, which affects query quality and ultimately leads to significant variance in average recall across different topics (clusters)? What would be the recall improvement of the proposed best belief pipeline in more challenging domains?
  - Lack of Case Analysis: There is no case study showing how backend models respond differently to different belief pipelines.
  - Limited Applicability: The proposed information retrieval approach seems only suitable for comprehensive integration tasks like survey papers. Would it be helpful for multi-hop QA tasks that require search?

---

> ### Author Response · Authors · 2025-12-03
> **Comment on Reviewer Z2dW's Comments**
>
> 1. **Regarding Weakness [2] (Limited Novelty).**
>    Our goal is not to emulate a fully open-ended environment such as the web. Instead, our intention is to introduce a controlled, modular workflow and a gym that enables systematic analysis of how variations in each module affect information-seeking performance. The primary contribution is thus the SeekerGym environment and its objectively measurable evaluation protocol, rather than a claim of novelty in the agent algorithms themselves.
>
> 2. **Regarding Questions [1,2] (Testing methodology and topic clusters).**
>    We group articles into clusters and use the same number of articles per cluster to maintain a balanced topic distribution. This design yields a more diverse and balanced dataset and also allows us to analyze how models perform across different topical areas. Some clusters involve more specialized terminology or more complex inter-paragraph dependencies, which makes them inherently more challenging. In addition, backend models may have varying familiarity with different domains due to their pretraining data, leading to heterogeneous performance across clusters. We agree that this phenomenon warrants deeper investigation, and in a revised version we plan to conduct a more detailed analysis of per-cluster difficulty and model behavior.
>
> 3. **Regarding Question [3] (Lack of case analysis).**
>    We appreciate this suggestion and agree that qualitative case analyses can clarify how different belief pipelines influence model behavior. In a revised version, we will include case studies illustrating how backend models respond under different belief representations and how these differences relate to the observed quantitative results.
>
> 4. **Regarding Question [4] (Applicability beyond survey-style tasks).**
>    Multi-hop QA benchmarks such as SimpleQA, BrowseComp, and HLE are primarily designed to find short, targeted answers, and thus emphasize memory management and noise filtering. In contrast, our task aims at completing comprehensive articles by modeling logical relationships between passages and generating queries to uncover missing information. While the Uncertainty-Informed Belief (UIB) mechanism could in principle be adapted to other benchmarks, its benefits would be more limited in settings where the objective is short-answer retrieval rather than holistic reconstruction. Our design is therefore tailored to comprehensive integration tasks (e.g., survey-style documents), where query planning and coverage are central.

---

### Official Review · Reviewer_mR8K · 2025-11-01

**Soundness:** 1
**Presentation:** 2
**Contribution:** 2
**Rating:** 4
**Confidence:** 3

**Summary:**

This work introduces SeekerGym, an environment for evaluation information-seeking agents. SeekerGym supports tasks constructed from two sources: wikipedia and CS survey papers. The metrics focuses on tracking the agents' abilities in reasoning about the exact missing information from trajectories. The author further propose SeekerAgent, featuring various belief structuring pipelines. Experiments show that the proposed pipelines effectively improves the performance of agents on SeekerGym.

**Strengths:**

1. This work contributes two environments for evaluating AI agents on challenging information-seeking tasks, which can be a meaningful resource for the community.
2. The proposed belief structuring pipelines seem to be effective scaffolding methods for enhancing agent performance.
3. Writing and presentation is overall clear.

**Weaknesses:**

1. I am not sure whether the use of "uncertainty" is properly used in the title. This work refers to "uncertainty" as what to search next at each step in the information-seeking process. This is quite different from the more commonly used definition for "uncertainty" in AI research, which is related to model confidence. If this is actually the intended use, I do not see a metric that quantifies the uncertainty of agents.
2. The metrics and reward is effectively tracking the change of recall of oracle documents in the information-seeking process. Which seems to be overly simple and can actually be used for any other deep research dataset with relevant documents/sources annotated. This potentially undermines the value of the SeekerGym as a separate benchmark.
3.  The proposed belief structuring pipeline method is somewhat interesting and effective. However, I think this method is generally applicable to the context management of any information-seeking agents. Therefore, I would like to see it tested on at least one existing benchmarks to concrete the methodological contribution.
4. The evaluation settings have issues. Firstly, why is each episode only evaluated with a single run instead of reporting average performance over multiple runs? Secondly, for RQ1, the conclusion regarding non-reasoning models perform better is not rigorous. I don't think comparing different models (Deepseek-R1/V3 and Gemini-2.0), which are trained very differently, under different inference setting (with and without reasoning) can lead to the conclusion.
5. Regarding the train/test split (line 213-215), why does splitting each cluster into train and test sets and train agents for each cluster can "facilitate learning"? Shouldn't a larger and more diverse training set benefit performance? Also, there doesn't seem to be details about how agents are trained in the paper.

**Questions:**

Please see weaknesses.

---

> ### Author Response · Authors · 2025-12-03
> **Authors' Comment on Reviewer Z2dW's Comments**
>
> 1. **Regarding the use of “uncertainty” in the title and method.**
>    We do not use “uncertainty” in the sense of probabilistic model confidence over predictions. Instead, our goal is to have the model generate **masks** over the current set of retrieved passages, indicating which parts of the belief state are still incomplete or insufficient for the downstream task. When these masks are incorporated into the belief representation (UIB), they improve the quality of subsequent query generation during information seeking by explicitly highlighting information gaps.
>
>    Concretely, the model is prompted to produce an “outspoken uncertainty” signal: given the current passages, it infers and marks (via masking) which aspects require further evidence or elaboration. This is closer to *expressing uncertainty through model inference over coverage* than to estimating numerical confidence. We will clarify this operational definition of “uncertainty” in the paper and, if needed, adjust the wording in the title and text to avoid confusion with standard probabilistic notions of uncertainty in AI.

---

### Author Response · Authors · 2025-12-03
**Author Response: SeekerGym Benchmark Clarifications**

We thank all reviewers for their constructive feedback. Below, we address several points that were raised consistently across reviews.

**Contribution and novelty of SeekerGym.** SeekerGym is a gym-based benchmark that addresses two key gaps in the existing literature on LLM capabilities and benchmarking.

First, **SeekerGym provides a multi-turn, agentic environment where the core metric of information-seeking is query generation**, rather than report generation. The benchmark focuses on a fundamental question:

> Does a research agent generate appropriate queries when gathering information?

We view query formulation as the critical capability in information-seeking, rather than the ability to produce plausible-sounding answers. Accordingly, our gym-based environment offers a modular, controlled setting that specifically measures querying capabilities in a multi-turn, retrieval-based context. LLM usage is strictly constrained to query generation (with the exception of the UIB module in the belief pipeline). Moreover, SeekerGym evaluates agents based on the aggregated set of retrieved paragraphs, rather than on generated reports (as in DeepResearchBench, USTC), so the reward is a robust measure that is not susceptible to biases inherent in LLM-as-judge evaluations.

Second, SeekerGym introduces a **recall-based measure of holistic information coverage** specific to each topic. We provide a recall metric that holistically captures how well an agent covers information for a given topic. In contrast to existing agentic information-seeking approaches that follow a "question-then-search-for-answer" pattern, we propose an **Article Reconstruction** framework (analogous to masking-based training) in which each topic–paragraphs pair is presented as a reconstruction task that requires appropriate query formulation.

To support this, we construct a gym with a dataset that faithfully represents the information coverage of specific topics (hence our use of survey papers and Wikipedia articles). The scalable dataset natively supported by SeekerGym comprises over 40,000 refined, high-quality, descriptive topic data points from 200 CS survey papers, organized more systematically than annotated or synthetic datasets (such as those in DeepResearchBench by FutureSearch). This credible ground truth provides a solid foundation for measuring LLM information-seeking capabilities.

**Comparison to existing benchmarks.** The table below situates SeekerGym relative to existing agent benchmarks.

Table 1: Comparison of agent benchmarks. Existing information-seeking benchmarks focus on either specific-scope tasks with ground-truth verification or holistic-scope tasks with LLM-based evaluation. SeekerGym addresses holistic information-seeking with ground-truth verification.

| Benchmarks                                                                  | Long horizon | Information scope | Verifier         | Task             |
| --------------------------------------------------------------------------- | -----------: | ----------------- | ---------------- | ---------------- |
| HotpotQA, MultihopQA, SimpleQA                                              |            ✗ | Specific          | Ground truth     | Info-seeking     |
| HLE, BrowseComp, xbench-DeepSearch                                          |            ✓ | Specific          | Ground truth     | Info-seeking     |
| SWE-Bench                                                                   |            ✗ | –                 | Ground truth     | Agentic          |
| SWE-Agent, WebArena, AppWorld, WebShop                                      |            ✓ | –                 | Ground truth     | Agentic          |
| DeepResearchGym, DeepResearchBench (USTC), DeepResearchBench (FutureSearch) |            ✓ | Holistic          | LLM-as-a-judge   | Info-seeking     |
| **SeekerGym (ours)**                                                        |        **✓** | **Holistic**      | **Ground truth** | **Info-seeking** |

(Continued in Second Comment)

---

### Author Response · Authors · 2025-12-03
**Author Response: Clarifications on SeekerAgent and Areas for Improvement**

First, we clarify that SeekerAgent (and its belief pipeline) is **not** intended to deliver state-of-the-art performance. Our primary contribution lies in SeekerGym as a controlled and objectively measurable benchmark; the SeekerAgent variants are provided as concrete instantiations to probe the benchmark rather than as proposed SOTA methods. The reported SeekerAgent algorithms serve three purposes:

1. to illustrate why even a basic belief pipeline (e.g., deduplication) is important,
2. to show how existing algorithms (e.g., Reflexion) behave in this agentic setting, and
3. to demonstrate that incorporating curiosity-inducing mechanisms into the belief representation (Uncertainty-Informed Belief) can yield substantial performance gains.

Across topics and models, we observe that different belief representations consistently affect the quality of LLM query generation. We view this persistent performance gap as evidence that SeekerGym captures a replicable and distinctive aspect of information-seeking—operationalized here as query generation—that is not adequately measured by existing benchmarks, largely due to their more complex and less controlled environments.

Second, we clarify the rationale for including the meta-reflection technique (inspired by Reflexion) and how the corresponding experiment is structured.

Cluster-based meta-information extraction. We group documents $w \in \mathcal{W}$ into semantic clusters $\mathcal{C} = {C_1, C_2, \ldots, C_K}$ using DBSCAN, where each cluster $C_j \subseteq \mathcal{W}$ corresponds to a coherent topic area. For evaluation, we select a subset of documents from each cluster to form the evaluation set
$\mathcal{W}{\text{eval}} = \bigcup{j=1}^K \mathcal{W}{\text{eval}}^{(j)}$,
where $\mathcal{W}{\text{eval}}^{(j)} \subset C_j$ are the evaluation documents from cluster $j$. The remaining documents
$\mathcal{W}{\text{held-out}}^{(j)} = C_j \setminus \mathcal{W}{\text{eval}}^{(j)}$
constitute a held-out set for each cluster.

For each cluster $C_j$, we extract meta-information—specifically, a topic schema—from the held-out documents $\mathcal{W}{\text{held-out}}^{(j)}$. This schema captures common organizational patterns and information categories that characterize documents in that cluster. During evaluation on a document $w \in \mathcal{W}{\text{eval}}^{(j)}$, the corresponding cluster-level meta-information is provided to the agent to guide its information-seeking strategy.

Connection to the Reflexion framework. The held-out set $\mathcal{W}{\text{held-out}}^{(j)}$ for each cluster can be viewed as providing demonstration experience analogous to the Reflexion framework. Because documents in $\mathcal{W}{\text{held-out}}^{(j)}$ are fully observable (all passages are available), they represent the endpoint of a complete information-seeking process in which the agent has successfully uncovered all relevant content. By analyzing these complete documents, we distill structural patterns that inform the agent’s beliefs about what information to seek when exploring new documents $w \in \mathcal{W}_{\text{eval}}^{(j)}$ from the same cluster. This serves as a form of prior knowledge that helps the agent navigate the partially observable environment more effectively.


**Areas for improvement.**

We acknowledge the following limitations and plan to address them in subsequent revisions:

- Related work. We will expand the related work section to cover additional benchmarks and methods in agentic information-seeking, and provide a more systematic discussion situating SeekerGym within existing approaches.
- Experimental rigor. We recognize the limitations of single-run evaluations and specific temperature settings. Future revisions will report results over multiple runs with appropriate statistical analysis to better assess robustness and reproducibility.
- Model coverage. We will broaden our evaluation to include a wider range of open-source models (e.g., Qwen, LLaMA) to better demonstrate the generality and applicability of our findings.

---

### Note · Authors · 2025-12-04

**Comment:**

We sincerely thank the reviewers and the area chairs for their time and feedback. After careful consideration, we have decided to withdraw this submission from further consideration at this venue.

**Withdrawal Confirmation:**

I have read and agree with the venue's withdrawal policy on behalf of myself and my co-authors.